# Effect of Local Adjuvants Following Curettage of Benign and Intermediate Tumours of Bone: A Systematic Review of the Literature

**DOI:** 10.3390/cancers15174258

**Published:** 2023-08-25

**Authors:** Maria Anna Smolle, Veronika Roessl, Andreas Leithner

**Affiliations:** Department of Orthopaedics and Trauma, Medical University of Graz, Auenbruggerplatz 5, 8036 Graz, Austria; veronika.roessl@stud.medunigraz.at (V.R.); andreas.leithner@medunigraz.at (A.L.)

**Keywords:** local adjuvant, benign bone tumour, intralesional resection, local recurrence, treatment

## Abstract

**Simple Summary:**

Benign and intermediate bone tumours are often treated by intralesional curettage together with local adjuvants in order to the reduce risk of local relapse. However, the role of different adjuvants used is still discussed controversially. In the present systematic literature review, altogether 3316 cases of benign/intermediate bone tumours were summarised with regards to the use of local adjuvants, as well as their respective impact on local recurrence. Overall, 32 different combinations of local adjuvants were identified. Although some tumour entities may benefit from the addition of a local adjuvant, the main treatment step remains thorough curettage of the lesion.

**Abstract:**

Local adjuvants are used upon intralesional resection of benign/intermediate bone tumours, aiming at reducing the local recurrence (LR) rate. However, it is under debate whether, when and which local adjuvants should be used. This PRISMA-guideline based systematic review aimed to analyse studies reporting on the role of adjuvants in benign/intermediate bone tumours. All original articles published between January 1995 and April 2020 were potentially eligible. Of 344 studies identified, 58 met the final inclusion criteria and were further analysed. Articles were screened for adjuvant and tumour type, follow-up period, surgical treatment, and development of LR. Differences in LR rates were analysed using chi-squared tests. Altogether, 3316 cases (10 different tumour entities) were analysed. Overall, 32 different therapeutic approaches were identified. The most common were curettage combined with high-speed burr (*n* = 774; 23.3%) and high-speed burr only (*n* = 620; 18.7%). The LR rate for studies with a minimum follow-up of 24 months (*n* = 30; 51.7%) was 12.5% (185/1483), with the highest rate found in GCT (16.7%; 144/861). In comparison to a combination of curettage, any adjuvant and PMMA, the sole application of curettage and high-speed burr (*p* = 0.015) reduced the LR rate in GCT. The overall complication rate was 9.6% (263/2732), which was most commonly attributable to postoperative fracture (*n* = 68) and osteoarthritis of an adjacent joint during follow-up (*n* = 62). A variety of adjuvants treatment options are reported in the literature. However, the most important step remains to be thorough curettage, ideally combined with high-speed burring.

## 1. Introduction

Benign and locally aggressive tumours and tumour-like lesions summarise a heterogeneous group of lesions preferably found in the bones of children, adolescents and young adults [1]. They are most frequently treated by intralesional curettage [2], sometimes combined with high-speed burring [3], as well as the adjunct of adjuvant treatments. The latter aim at further reducing the local recurrence (LR) rate. Phenol is used as an adjuvant as it leads to chemical coagulation of protein-rich substances and thus necrosis of remnant tumour cells [4]. 

Cryotherapy with liquid nitrogen can likewise destroy remnant tumour cells not removed during curettage but is also associated with complications such as delayed bone healing, wound-healing deficits, and fractures [5,6]. Ethanol is used to wash out phenol, but it also acts as a local adjuvant by exerting cytotoxic effects, yet to a lesser extent than phenol [7]. Further adjuvants used following bone tumour curettage include liquid zinc chloride [8], hydrogen peroxide [9], and argon-plasma laser [10].

Polymethylmethacrylate (PMMA) is not only considered an adjuvant for having thermal properties during polymerisation thus eventually leading to necrosis of remnant tumour cells, but it is also a defect filler [11]. Furthermore, its radio-opaque nature enables early detection of potential LRs [12,13]. However, there is an ongoing debate regarding whether, when and which adjuvants should be applied for different benign and locally aggressive tumour entities, given their differing aggressiveness, side effects and potential to locally recur [14].

The present systematic literature review aimed at analysing treatment combinations with adjuvants beyond surgical therapy (i.e., curettage and/or high-speed burr) of benign and locally aggressive tumours as well as tumour-like lesions. Furthermore, the frequency of local recurrences and complications associated with the different adjuvant treatments were investigated.

## 2. Materials and Methods

A systematic literature review in PubMed, adhering to the Preferred Reporting Items for Systematic Reviews and Meta-Analyses (PRISMA) guidelines, was performed. All English, Spanish or German original articles published from January 1995 to April 2020 and dealing with the application of adjuvants in benign and semi-malignant tumours of bone were potentially eligible. 

The following search terms were used: adjuvant treatment AND curettage AND benign bone tumour; adjuvant treatment AND curettage AND intermediate bone tumour; adjuvants AND curettage AND benign bone tumour; adjuvants AND curettage AND intermediate bone tumour; curettage AND adjuvant treatment AND benign bone tumour; curettage AND adjuvant treatment AND intermediate bone tumour (Appendix A).

Of the initially 281 studies identified through the defined search terms and 63 additional articles selected through cross-reference checking, 171 duplicates were removed. The remaining 173 articles were subsequently screened. In the next step, 27 articles were excluded for not meeting the inclusion criteria based on the title and/or abstract, leaving 146 studies to be assessed for eligibility. Of these, 35 were excluded for not being in the English, German, or Spanish language, or for lacking complete information on the adjuvant treatment applied (*n* = 53). Therefore, 58 studies could be included in the final analysis, of which 30 reported on patients with a minimum follow-up ≥ 24 months, and 28 reported on patients with a minimum follow-up less than 24 months (Figure 1). Screening and initial checks for the eligibility of the studies were performed by one author (V.R.), and re-checking of these potential studies was performed by a second author (M.A.S.).

The following information was ascertained for each study included: study type, title, author, year of publication, journal, time of follow-up (mean; minimum, maximum), histological diagnosis, number of patients/tumours treated, type of surgical therapy (curettage, high-speed burr), type of adjuvant used (phenol, liquid nitrogen [cryotherapy], ethanol, electrocauterization, argon-plasma coagulation, hydrogen peroxide [H_2_O_2_], PMMA, sodium hyponitrite [Na_2_N_2_O_2_], liquid zinc chloride), type of defect filling (autograft, allograft, synthetic device), number of cases with complications, type of complications, treatment groups affected by complications, and number of cases with local recurrence. Notably, PMMA was considered an active adjuvant (due to its thermic properties) rather than a defect filling material. 

In case of incomplete case-based information on treatment combinations and/or recurrence rates applied within individual studies, only those patients with full information available were included in the systematic review. As some studies did report on more than one tumour in a single patient, tumour cases rather than individual patients were counted.

### Statistical Analysis

Descriptive and explorative analyses were performed. To allow for evaluation of local recurrence beyond 2 years following surgery, separate analyses for studies reporting on follow-up less than (*n* = 28) or beyond 24 months (*n* = 30) were performed. For the largest 4 tumour entities (giant cell tumour of bone [GCT], aneurysmal bone cyst [ABC], atypical cartilaginous tumour [ACT], chondroblastoma), LR rate was assessed depending on adjuvant treatments applied (for studies with a minimum follow-up ≥ 24 months only) and estimated using a chi-squared or Fisher’s exact test. Curettage combined with high-speed burr was defined as the control group for the individual treatments. For better comparison, treatment groups were summarised as follows: curettage only, curettage + burr, curettage + adjuvant, curettage + PMMA, curettage + burr + adjuvant, curettage + adjuvant + PMMA, curettage + burr + PMMA, curettage + burr + adjuvant + PMMA. As not all studies provided information on complications, relative frequencies of complications were calculated based on the patient number with information on complications available (*n* = 2732). A *p*-value of < 0.05 was considered statistically significant. All statistical analyses were carried out using Stata Version 16.1 (StataCorp, College Station, TX, USA).

## 3. Results

A total of 3316 cases with 10 different tumours and tumour-like lesions were reported in the 58 studies [7,8,9,10,12,13,15,16,17,18,19,20,21,22,23,24,25,26,27,28,29,30,31,32,33,34,35,36,37,38,39,40,41,42,43,44,45,46,47,48,49,50,51,52,53,54,55,56,57,58,59,60,61,62,63,64,65]. The most common entities were GCT in 2235 cases (67.4%), ACT in 333 cases (10.0%), ABC in 262 cases (7.9%), enchondroma in 235 cases (7.1%), and chondroblastoma in 219 cases (6.6%; Table 1).

After excluding studies reporting on follow-up less than 24 months, 1483 cases in 30 studies remained in the analysis [8,9,12,15,16,17,18,20,21,27,33,35,37,40,41,42,43,46,48,50,52,53,54,55,57,58,61,63,64,65], with GCT again contributing to the majority of cases (*n* = 861; 58.1%; Table 1).

### 3.1. Treatment

Altogether, 32 different therapeutic approaches were identified in the 58 studies (Appendix A). The most common treatment was curettage combined with high-speed burr (*n* = 774; 23.3%), followed by curettage only (*n* = 620; 18.7%), and curettage combined with PMMA (*n* = 243; 7.3%). Defect filling (apart from PMMA) was carried out in 1390 of 3316 cases identified. The vast majority had received bone transplants (*n* = 1357; 97.6%), and bioactive materials and synthetic bone transplants had been used in 29 (2.1%) and 4 cases (0.3%), respectively. In detail, defect filling devices involved an auto- or allograft (as not further specified by studies; *n* = 615), a bone transplant (without further specification; *n* = 408), an autograft (*n* = 153), an allograft (*n* = 125), a spongious bone transplant (*n* = 56), bioactive materials (*n* = 29), synthetic bone transplants (*n* = 3), or hydroxyapatite (*n* = 1).

#### 3.1.1. GCT

Altogether, 22 different treatment combinations were identified for GCT (*n* = 2235). Curettage combined with high-speed burr was the most frequent one (*n* = 665; 29.8%), followed by curettage only (*n* = 272; 12.2%) and curettage combined with PMMA (*n* = 229; 10.2%).

#### 3.1.2. ACT

Fourteen different treatment combinations were reported for ACT. Of the 333 ACT cases, 93 were treated with a combination of curettage, phenol and PMMA (*n* = 93; 27.9%). Furthermore, 85 cases had received curettage, phenol and ethanol (25.5%), and 56 cases received curettage only (16.8%).

#### 3.1.3. ABC

Ten different treatments were identified for ABC (*n* = 262). A combination of curettage, high-speed burr and cryotherapy was the most common (*n* = 98; 37.4%). A further 41 cases had undergone curettage and high-speed burr only (15.6%), and 31 cases had undergone a combination of curettage, high-speed burr, ethanol and electrocauterization (11.8%).

#### 3.1.4. Chondroblastoma

The 219 chondroblastoma cases had been treated with 11 differing combinations. The most frequent ones were curettage combined with high-speed burr (*n* = 97; 44.3%) and curettage only (*n* = 47; 21.5%).

#### 3.1.5. Enchondroma

Four treatment combinations were identified for the 235 enchondromas. The vast majority had undergone curettage only (*n* = 189; 80.4%). Sixteen cases had been treated with a combination of curettage, high-speed burr, H_2_O_2_ and PMMA (6.8%), and another 16 cases had been treated with curettage, high-speed burr, cryotherapy and PMMA (6.8%). The remaining 14 cases had undergone curettage, cryotherapy and sodium hyponitrite treatment (6.0%).

#### 3.1.6. Rare Tumours/Tumour-like Lesions

Of the seven osteoblastomas identified, three had been treated with curettage, high-speed burr and phenol, two with curettage only, one with curettage, high-speed burr and cryotherapy, and one with curettage and PMMA. Curettage only had been used in all fibrous dysplasia (*n* = 14), osteoid osteoma (*n* = 5), NOF (*n* = 5), and chondromyxoid fibroma (*n* = 1) cases.

### 3.2. Local Recurrences

Overall, 602 LRs were reported in 3316 cases, amounting to an overall recurrence rate of 18.2%. Of 1483 cases with a minimum follow-up of 24 months, 185 developed a local recurrence (12.5%). Thereafter, all LR-related analyses were carried out for cases with a minimum follow-up of 24 months.

Split by the four most frequent tumour entities, LR rates amounted to 16.7% for GCT (144/861), 9.0% for chondroblastoma (18/201), 8.5% for ABC (14/165), and 1.2% for ACT (2/167).

The remaining five LRs had been reported in fibrous dysplasia (2/14), osteoid osteoma (2/5), and NOF (1/5).

For GCT, LR rate could be significantly reduced by a combination of curettage and high-speed burr in comparison to curettage, adjuvant and PMMA (28.9%; *p* = 0.015). LR rates for the remaining treatment combinations were not significantly different in comparison to curettage and high-speed burr (all *p* > 0.05; Table 2).

No significant difference in LR rate could be found for any treatment combination compared to curettage and high-speed burr for the treatment of ABC (all *p* > 0.05; Table 3).

Likewise, for ACT, none of the treatment combinations significantly altered the LR rate in comparison to the defined “standard treatment” of curettage and high-speed burr (all *p* > 0.05 or *p*-value not calculated; Table 2).

Curettage combined with adjuvant and PMMA in chondroblastoma performed worse in terms of LR rate (100%) as compared with curettage and high-speed burr (8.2%; *p* = 0.009). All other treatment combinations reached similar LR rates to curettage and high-speed burr only (all *p* > 0.05; Table 2).

### 3.3. Complications

Nine of fifty-eight studies (involving 584 patients) did not provide information on any complications [13,19,28,35,36,39,44,47,62]. A further nine studies reported 0 complications in their respective cohorts (altogether 436 patients) [17,21,23,24,32,43,49,54,63]. In the remaining 40 studies (involving 2296 patients), at least one complication had been observed [7,8,9,10,12,15,16,18,20,25,26,27,29,30,31,33,34,37,38,40,41,42,45,46,48,50,51,52,53,55,57,58,59,60,61,64,65,66,67,68]. Thus, 263 complications occurred in 2732 patients with information on adverse events available, amounting to an overall complication rate of 9.6%. The most common complications were postoperative fractures (*n* = 68/2732, 2.5%), osteoarthritis of the adjacent joint during follow-up (*n* = 62/2732; 2.3%), persisting pain (*n* = 30/2732; 1.1%), and nerve injuries (*n* = 14/2732; 0.5%). A detailed description of complications, frequencies and affected treatment group is visible in Table 3.

The highest complication rate was present in the cohort of patients treated with curettage + high-speed burr + argon beam + H_2_O_2_ + PMMA (14/38; 36.8%), followed by curettage + cryotherapy (11/28; 28.2%), curettage + high-speed burr + H_2_O_2_ + PMMA (6/29; 20.7%), curettage + phenol (1/5; 20.0%), curettage + ethanol (2/12; 16.7%), and curettage + cryotherapy + sodium hyponitrite (4/26; 15.4%). Figure 2 provides detailed information on complications per treatment group together with relative percentages.

## 4. Discussion

According to the present systematic review, 32 different treatment combinations have been described in the literature for curettage of benign and locally aggressive tumours of bone, as well as tumour-like lesions. Most lesions (with GCT of bone, followed by ACT and ABC being the most common entities) were treated with curettage and high-speed burr, or curettage only. Overall LR and complication rates amounted to 18.2% (12.5% in studies with a minimum follow-up of 24 months) and 9.6%.

GCT of bone presented with the highest LR rate, which appeared to be reduced by the addition of phenol and PMMA in comparison to curettage and high-speed burr alone. On the other hand, chondroblastoma LR rates were reduced by the addition of high-speed burr rather than any further adjuvant.

Intralesional curettage of benign bone tumours and tumour-like lesions aims at reducing the rate of LR to a minimum. This goal is achieved by meticulous curettage (i.e., macroscopic removal) of the tumour either alone or in combination with high-speed burr to also target the lesion’s border by mechanical disruption [61,65,69]. However, differing tumour biology also results in varying risk of recurrence that must be taken into consideration upon surgery and in the decision to use adjuvants and which to use.

Unsurprisingly, GCT of bone had the highest local recurrence rate, amounting to 16.7% for studies with a minimum follow-up of 24 months. Interestingly, the sole use of curettage and high-speed burr appeared to significantly reduce the LR rate in comparison to curettage, use of an adjuvant, and PMMA. This contradicts observations made by Gava et al. [70] in a systematic review mainly focussing on defect filling rather than the different adjuvant treatments, which discovered that the LR rate tends to be reduced in GCT upon use of an adjuvant. On the other hand, the systematic review and meta-analysis by Algawahmed et al. reached the conclusion that the use of adjuvants does not significantly alter the LR rate in GCT [14].

In chondroblastoma, the addition of high-speed burr to curettage of the lesion seems to be of greater importance in terms of LR-rate reduction than any adjuvant applied, yet the low number of cases reporting on treatment combinations omitting high-speed burring has to be considered when interpreting these results [16,57].

The cumulative analysis of adjuvant treatments for ACTs of the extremities, formerly known as chondrosarcoma G1, did not allow identification of any adjuvant that would significantly reduce LR rate. This was mainly caused by the overall low LR rate reported (1.2%; 2/167) that was even lower than the one identified for their benign counterpart enchondroma (3.5%; 2/57). This fact appears surprising at first, given that ACTs are considered more aggressive than EC, with higher LR rates [71]. Yet, the numbers should be interpreted with caution, given that the diagnostic criteria to distinguish between enchondroma and ACT are yet to be defined and may have varied between studies and time periods [72,73].

ABCs were historically considered tumour-like lesions emerging due to increased intraosseous venous pressure. However, the detection of a ubiquitin-specific protease (*USP6*) gene translocation within some ABCs has moved them towards the group of “true” neoplasms [74]. According to this systematic review, no specific adjuvant seems to significantly alter LR rate in comparison to curettage and high-speed burr alone, which is also in line with conclusions reached by other authors [58,69]. Notably, percutaneous techniques with sclerotherapy are becoming more common in the treatment of ABCs, with the main advantage being low morbidity and the possibility to perform repeated procedures [75].

Interestingly, a study reporting on curettage of NOF as a tumour-like lesion (alongside other benign bone tumours) was likewise identified in this systematic review, although NOFs are nowadays considered as “leave-me-alone” lesions [46].

Finally, potential side effects of adjuvants reported in the literature and/or still applied in clinical practice must be taken into consideration. The overall complication rate herein observed amounted to 9.6%, with postoperative fractures and osteoarthritis of the adjacent joint during follow-up being the most common. For example, phenol is known for its toxic properties when being inhaled, ingested or in contact with the skin [68]. The latter leads to bleaching and irritation, whereas signs of phenol intoxication following ingestion include nausea, pain, and diarrhoea [76]. Ultimately, intoxication may lead to renal and hepatic damage, CNS depression, pulmonary oedema and circulatory as well as respiratory failure [68,76]. Therefore, phenol is nowadays only rarely used as an adjuvant, and has even been prohibited in some countries for safety reasons [66]. For other adjuvants such as PMMA, no such toxic side effects have been reported, either for patients or surgeons, rendering them a safe and effective agent. Interestingly, the treatment combination most commonly associated with complications was curettage combined with high-speed burr, argon beam, H_2_O_2_ and PMMA (36.8% complication rate), followed by curettage and cryotherapy (28.2% complication rate). The latter rate is not surprising, given the fact that the application of cryotherapy has been reported to lead to local tissue necrosis resulting in early (e.g., skin necrosis) or late (e.g., postoperative fracture) complications [5,77,78]. However, one has to be cautious with the provided numbers, as not every study reported on complications, and some did not provide sufficient information to allow delineation regarding which treatment group the respective adverse event had occurred.

Some limitations have to be considered when interpreting the results of the present study. The heterogeneity of studies included (10 different tumour entities, 32 various treatment combinations) impeded a large and uniform analysis, specifically regarding the impact of adjuvant-combinations on LR rate. Also, detailed descriptions of treatment combinations used by the authors and the resulting LR rates for individual patients were not uniformly available; therefore, on some occasions not all patients reported within an individual study could be included. This has to be considered as another limitation of the study, which eventually distorted the influence of specific adjuvants on cumulative LR rate. Another limitation is the fact that nearly half of the studies had to be excluded when calculating the LR rate, given that the minimum follow-up of 24 months had not been reached. Furthermore, as already outlined above, not every study provided sufficient information on the occurrence of potential complications, or allowed for delineation regarding which treatment group had ultimately been affected by the respective complication. Thus, the given complication numbers have to be interpreted bearing these limiting factors in mind.

## 5. Conclusions

In conclusion, the present systematic review retrieved a myriad of treatment combinations used upon curettage of benign and locally aggressive bone tumours as well as tumour-like lesions. Despite these available treatment options, the overall most important treatment to lower the rate of recurrence appears to be meticulous surgical curettage, either alone or preferably in combination with high-speed burring.

## Figures and Tables

**Figure 1 cancers-15-04258-f001:**
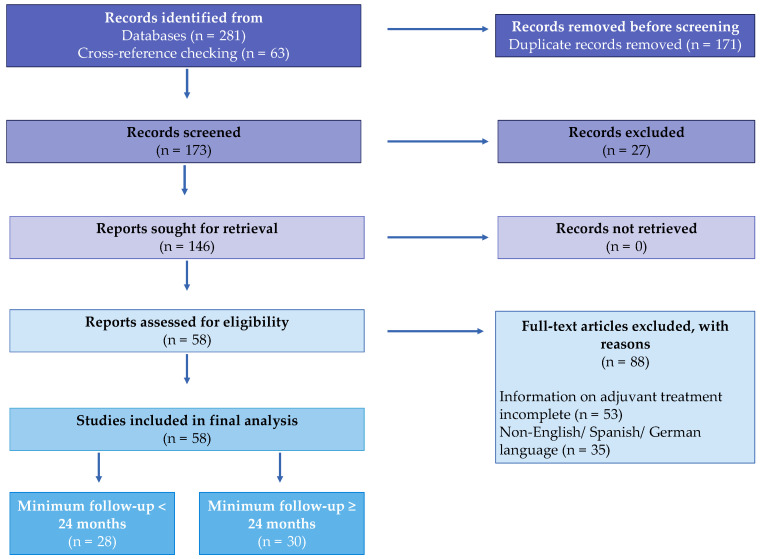
PRISMA flow chart showing selection of studies included in the systematic review.

**Figure 2 cancers-15-04258-f002:**
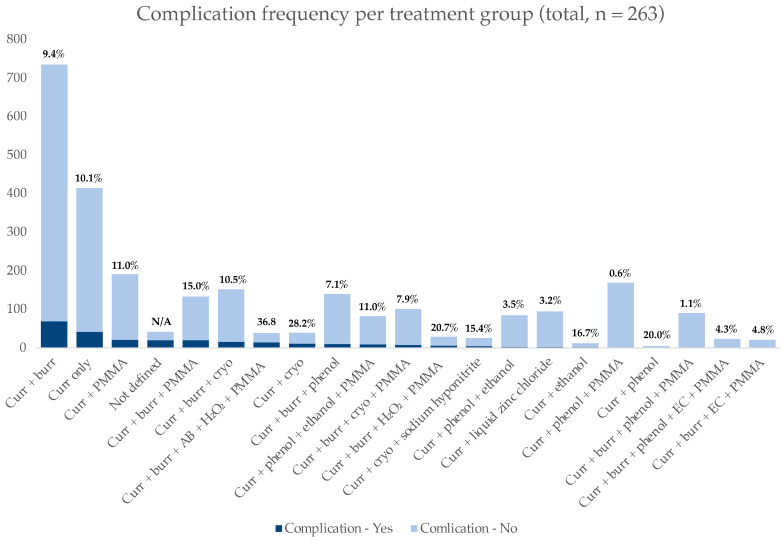
Frequency of complications as reported by treatment group. Relative percentages of complication frequencies per treatment group are provided. Legend: Curr—curettage; AB—argon beam; cryo—cryotherapy; EC—electrocauterisation.

**Table 1 cancers-15-04258-t001:** Number of cases separated by tumour entities and minimum follow-up less than or more than 24 months.

Tumour Entity	Total Case Number (*n* = 3316)	Minimum Follow-Up ≥ 24 Months (*n* = 1483)	Minimum Follow-Up < 24 Months (*n* = 1833)
GCT	2235	861	1374
Chondroblastoma	219	201	18
ABC	262	165	97
ACT	333	167	166
EC	235	57	178
Osteoblastoma	7	7	0
Fibrous dysplasia	14	14	0
Non-ossifying fibroma	5	5	0
Osteoid osteoma	5	5	0
Chondromyxoid fibroma	1	1	0

Legend: GCT—giant cell tumour of bone; ABC—aneurysmal bone cyst; ACT—atypical cartilaginous tumour; EC—enchondroma.

**Table 2 cancers-15-04258-t002:** Impact of adjuvant treatments on local recurrence rate for studies with a minimum follow-up of 24 months, split by the 4 most common tumour entities.

**Giant Cell Tumour (GCT) of Bone**				
	**Total**	**No LR**	**LR**	** *p* ** **-value**
**Curettage + burr**	**263**	**219**	**44**	**N/A**
Curettage	12	8	4	0.233 *
Curettage + adjuvant	95	83	12	0.346
Curettage + adjuvant + PMMA	83	59	24	**0.015**
Curettage + burr + adjuvant	164	133	31	0.602
Curettage + burr + adjuvant + PMMA	223	198	25	0.082
Curettage + burr + PMMA	22	18	4	0.773 *
Curettage + PMMA	1	1	0	n.c.
**Aneurysmal Bone Cyst (ABC)**				
	**Total**	**No LR**	**LR**	** *p* ** **-value**
**Curettage + burr**	**36**	**34**	**2**	**N/A**
Curettage	20	19	1	1.000 *
Curettage + burr + adjuvant	101	91	10	0.428
Curettage + burr + adjuvant + PMMA	2	2	0	1.000 *
Curettage + PMMA	6	5	1	0.378 *
**Atypical Cartilaginous Tumour (ACT)**				
	**Total**	**No LR**	**LR**	** *p* ** **-value**
**Curettage + burr**	**5**	**0**	**0**	**N/A**
Curettage	7	6	1	1.000 *
Curettage + adjuvant	23	23	0	n.c.
Curettage + adjuvant + PMMA	93	0	0	n.c.
Curettage + burr + adjuvant	16	15	1	1.000 *
Curettage + burr + adjuvant + PMMA	21	21	0	n.c.
Curettage + burr + PMMA	1	1	0	n.c.
Curettage + PMMA	1	0	0	n.c.
**Chondroblastoma**				
	**Total**	**No LR**	**LR**	** *p* ** **-value**
**Curettage + burr**	**97**	**89**	**8**	**N/A**
Curettage	34	30	4	0.508 *
Curettage + adjuvant + PMMA	2	0	2	**0.009 ***
Curettage + burr + adjuvant	34	32	2	1.000 *
Curettage + burr + PMMA	28	28	0	0.116
Curettage + PMMA	6	4	2	0.103 *

Legend: * Fisher’s exact test; N/A—not applicable; n.c.—not calculated. *p*-values in bold highlight statistically significant results.

**Table 3 cancers-15-04258-t003:** Type of complications reported in individual studies, together with frequencies and treatment groups affected.

Type of Complication	Count (*n*; % of 263)	Affected Treatment Group
Postoperative fracture	68 (25.9%)	Curettage + burr (*n* = 24) Curettage + burr + argon beam + H_2_O_2_ + PMMA (*n* = 6) Curettage only (*n* = 5) Curettage + burr + PMMA (*n* = 5) Curettage + burr + H_2_O_2_ + PMMA (*n* = 4) Curettage + burr + cryotherapy + PMMA (*n* = 3) Curettage + burr + cryotherapy (*n* = 2) Curettage + burr + phenol (*n* = 2) Curettage + phenol + ethanol (*n* = 2) Curettage + cryotherapy + sodium hyponitrite (*n* = 2) Curettage + cryotherapy (*n* = 1) Curettage + ethanol (*n* = 1) Not defined (*n* = 11)
Osteoarthritis of adjacent joint	62 (23.6%)	Curettage + burr (*n* = 19) Curettage + burr + PMMA (*n* = 12) Curettage + burr + argon beam + H_2_O_2_ + PMMA (*n* = 8) Curettage + phenol + ethanol + PMMA (*n* = 6) Curettage + PMMA (*n* = 4) Curettage + burr + cryotherapy + PMMA (*n* = 4) Curettage + burr + cryotherapy (*n* = 3) Curettage only (*n* = 2) Curettage + liquid zinc chloride (*n* = 2) Curettage + burr + phenol + PMMA (*n* = 1) Curettage + burr + phenol + electrocauterization + PMMA (*n* = 1)
Persisting pain	30 (11.4%)	Curettage only (*n* = 21)
		Curettage + PMMA (*n* = 7)
		Curettage + burr (*n* = 2)
Deep wound infection	16 (6.1%)	Curettage + burr + cryotherapy (*n* = 3) Curettage only (*n* = 2) Curettage + cryotherapy (*n* = 2) Curettage + burr + PMMA (*n* = 2) Curettage + burr (*n* = 1) Curettage + liquid zinc chloride (*n* = 1) Curettage + burr + phenol (*n* = 1) Curettage + cryotherapy + sodium hyponitrite (*n* = 1) Curettage + phenol + ethanol + PMMA (*n* = 1) Curettage + burr + cryotherapy + PMMA (*n* = 1) Not defined (*n* = 1)
Nerve injury	14 (5.3%)	Curettage + cryotherapy (*n* = 4)
		Curettage + burr + cryotherapy (*n* = 4)
		Curettage only (*n* = 2)
		Curettage + burr (*n* = 1)
		Curettage + PMMA (*n* = 1)
		Curettage + burr + PMMA (*n* = 1)
		Curettage + burr + phenol (*n* = 1)
Superficial wound infection	12 (4.6%)	Curettage + burr (*n* = 3)
		Curettage only (*n* = 2)
		Curettage + ethanol (*n* = 1)
		Curettage + burr + cryotherapy (*n* = 1)
		Curettage + burr + phenol (*n* = 1)
		Curettage + phenol + ethanol (*n* = 1)
		Curettage + burr + electrocauterization + PMMA (*n* = 1)
		Not defined (*n* = 2)
Restricted mobility	10 (3.8%)	Curettage + burr (*n* = 5)
		Curettage + PMMA (*n* = 2)
		Curettage only (*n* = 1)
		Curettage + burr + phenol (*n* = 1)
		Not defined (*n* = 1)
Physeal arrest	9 (3.4%)	Curettage only (*n* = 2)
		Curettage + burr (*n* = 2)
		Curettage + burr + phenol (*n* = 2)
		Curettage + PMMA (*n* = 1)
		Curettage + burr + cryotherapy (*n* = 1)
		Not defined (*n* = 1)
Joint collapse	6 (2.3%)	Curettage + burr (*n* = 3)
		Curettage + cryotherapy (*n* = 3)
Limb deformity	5 (1.9%)	Curettage + burr (*n* = 2)
		Curettage + PMMA (*n* = 2)
		Not defined (*n* = 1)
Non-union	5 (1.9%)	Curettage + burr (*n* = 3)
		Curettage + phenol + ethanol + PMMA (*n* = 2)
Implant irritation	4 (1.5%)	Curettage + burr (*n* = 2)
		Curettage + burr + H_2_O_2_ + PMMA (*n* = 2)
Skin necrosis	3 (1.1%)	Curettage only (*n* = 1)
		Curettage + cryotherapy (*n* = 1)
		Not defined (*n* = 1)
Crack of affected bone	2 (0.8%)	Not defined (*n* = 2)
Delayed wound healing	2 (0.8%)	Curettage only (*n* = 2)
Abnormal banded signal around PMMA on MRI	2 (0.8%)	Curettage + PMMA (*n* = 2)
Intraoperative fracture	2 (0.8%)	Curettage + burr (*n* = 1)
		Curettage + phenol (*n* = 1)
PMMA excavation	2 (0.8%)	Curettage + PMMA (*n* = 1)
		Curettage + phenol + PMMA (*n* = 1)
Cartilage defect	1 (<0.5%)	Curettage + burr + cryotherapy (*n* = 1)
Deep vein thrombosis	1 (<0.5%)	Curettage only (*n* = 1)
Lung embolism	1 (<0.5%)	Curettage only (*n* = 1)
Unknown	1 (<0.5%)	Curettage + burr + phenol (*n* = 1)
Periarticular ossification	1 (<0.5%)	Curettage + burr (*n* = 1)
Bone graft reabsorption	1 (<0.5%)	Curettage + burr + phenol (*n* = 1)
Secondary sarcoma	1 (<0.5%)	Curettage + PMMA (*n* = 1)
Skin blisters	1 (<0.5%)	Curettage + burr + cryotherapy (*n* = 1)
Venous gas embolism	1 (<0.5%)	Curettage + cryotherapy + sodium hyponitrite (*n* = 1)

## Data Availability

The data presented in this study are available on request from the corresponding author.

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
