# Peer review of "Effect of Local Adjuvants Following Curettage of Benign and Intermediate Tumours of Bone: A Systematic Review of the Literature"

_cancers, 2023, doi:10.3390/cancers15174258_

Round 1

Reviewer 1 Report (New Reviewer)

The authors sincerely responded to the reviewers' comments, and I believe this paper deserves acceptance.

Author Response

Thank you for your valuable time and the positive appraisal of the manuscript.

Reviewer 2 Report (New Reviewer)

In my view, that authors responded to all comments raised by the previous reviewers and as such the present manuscript could be accepted for publication. One discretional comment In line 295 "H2O2 and PMMA (36.8% complication rate), followed by curettage and" Should be with subscript number 2 in both positions for hydrogenperoxyde

Author Response

The authors would like to thank the reviewer for their valuable time and the overall positive appraisal of the manuscript. We do apologise for this typo that has been corrected accordingly.

This manuscript is a resubmission of an earlier submission. The following is a list of the peer review reports and author responses from that submission.

Round 1

Reviewer 1 Report

One can appreciate the goals of this paper in its thorough systematic literature review of benign and intermediate bone tumors. However, this paper's main conclusion that adequate surgical curettage ± burring for all lesions is the mainstay tactic to decrease local recurrence rates is not a new or significant finding, but has been widely accepted as first line treatment. Furthermore, the statistical comparisons between different treatment combinations is widely imbalanced in group numbers, making it difficult to see the value in the statistically significant findings reported. It may be better to group the treatment modalities in categories, such as curettage only, curettage + burr only, curettage + burr + adjuvant, curettage + PMMA ± adjuvant, and so forth. Furthermore, it would be more interesting in terms of contributing more novel ideas to include other complications involved with the other adjuvants. Additionally, reconsideration of which bone tumors are included in this study may be necessary, as some of the tumors included have vastly different locally aggressive behavior, skewing the data. With fewer than 50% of review articles included having adequate follow up of 24 months or more, the discrepancies between the methods and findings are more apparent. 

In terms of the paper's writing, the message was adequately delineated. However, the paper had a minor set of awkward phrasing and/or grammatically inappropriate terms. As an example, in line 74 and 75, "as not" should be replaced with "for not".  

Reviewer 2 Report

The authors present a very thorough analysis of "the ink bottle" of adjuvants in the treatment of benign bone tumors. The study seems very sound and the subject is very interesting.

It is furthermore interesting that it is focused on local treatment and does not involve systemic treatment of any kind.

The results are very clearly presented and caution is advised when necessary i.e. regarding ACT.

I do not have any specific questions.

My sole recommendation would be to propose table 2 as supplementary material as it is quite large and does not bring much to support the text.

Reviewer 3 Report

An important systematic review in a issue of interest for all onco orthopedic centers.